# Harnessing Large Language Models for Text-Rich Sequential Recommendation

## ABSTRACT

Recent advances in Large Language Models (LLMs) have been changing the paradigm of Recommender Systems (RS). However, when items in the recommendation scenarios contain rich textual information, such as product descriptions in online shopping or news headlines on social media, LLMs require longer texts to comprehensively depict the historical user behavior sequence. This poses significant challenges to LLM-based recommenders, such as over-length limitations, extensive time and space overheads, and suboptimal model performance. To this end, in this paper, we design a novel framework for harnessing Large Language Models for Text-Rich Sequential Recommendation (LLM-TRSR). Specifically, we first propose to segment the user historical behaviors and subsequently employ an LLM-based summarizer for summarizing these user behavior blocks. Particularly, drawing inspiration from the successful application of Convolutional Neural Networks (CNN) and Recurrent Neural Networks (RNN) models in user modeling, we introduce two unique summarization techniques in this paper, respectively hierarchical summarization and recurrent summarization. Then, we construct a prompt text encompassing the user preference summary, recent user interactions, and candidate item information into an LLM-based recommender, which is subsequently fine-tuned using Supervised Fine-Tuning (SFT) techniques to yield our final recommendation model. We also use Low-Rank Adaptation (LoRA) for Parameter-Efficient Fine-Tuning (PEFT). We conduct experiments on two public datasets, and the results clearly demonstrate the effectiveness of our approach.

## KEYWORDS

Recommender System, Large Language Model, Sequential Recommendation

**ACM Reference Format:**

Anonymous Author(s). 2018. Harnessing Large Language Models for Text-Rich Sequential Recommendation. In *Proceedings of Make sure to enter the correct conference title from your rights confirmation emai (Conference acronym 'XX).* ACM, New York, NY, USA, 9 pages. https://doi.org/XXXXXXX.XXXXXXX

## 1 INTRODUCTION

Recently, Large Language Models (LLM), exemplified by ChatGPT[1], have demonstrated remarkable capabilities in the field of Natural

[1]https://chat.openai.com/

Language Processing (NLP), capturing the attention of numerous researchers. Owing to the strong reasoning and zero/few-shot learning capabilities exhibited by LLMs, many researchers are also exploring their application in other domains, such as Recommender Systems (RS) [30]. According to Wu et al. [30], a typical paradigm for employing LLM as RS involves feeding user profiles, behavioral data, and task instruction into the model, with the expectation that the LLM will offer a reasonable recommendation result in return. For example, Bao et al. [2] propose TALLRec, which converts the history sequence and new item to "Rec Instruction" and "Rec Input" as the input for the LLM model.

However, in recommendation scenarios where items have rich textual information, e.g., product titles in e-commerce, news headlines on media platforms, extended text becomes essential to comprehensively depict a user historical behavior sequence, which introduces the following challenges to LLMs. First, existing LLMs typically impose limitations on the length of the input, e.g., 1,024 tokens for GPT-2 [22], which may be insufficient to encompass extensive textual information. Second, due to the $O(n^2)$ computational complexity of the Transformer [27] architecture, prolonged texts lead to significant computational resource overheads for downstream recommendation tasks, which poses challenges to applications of recommender systems that demand high real-time responsiveness. Third, lengthier texts can make it more challenging for the model to effectively capture shifts in user preferences, potentially hindering optimal performance [14].

To this end, in this paper, we design a novel framework for harnessing Large Language Models for Text-Rich Sequential Recommendation (LLM-TRSR). Figure 1 shows a schematic diagram of our proposed method. Specifically, our method mainly consists of the following primary steps. Initially, we extract the user behavioral history sequence and transform it into an extended piece of text. Subsequently, this long text is segmented into several blocks, ensuring that each block can be fully ingested by large language models. We then propose an LLM-based summarizer that holistically considers these blocks to derive a summary of user preference. Note that the parameters of this summarizer are frozen. Finally, we build the input prompt text based on the user preference summary, recent user interactions, and candidate item information, and feed the prompt into an LLM-based recommender, which is subsequently fine-tuned using Supervised Fine-Tuning (SFT) to output "Yes" or "No". Additionally, a Parameter-Efficient Fine-Tuning (PEFT) method based on Low-Rank Adaptation (LoRA) to reduce memory overhead and expedite the training process.

In the aforementioned process, a pivotal question is how to employ an LLM-based summarizer to extract user preference from multiple blocks of user behavior. In this paper, inspired by two neural network architectures which are extensively applied in the deep learning domain, respectively Convolutional Neural Network

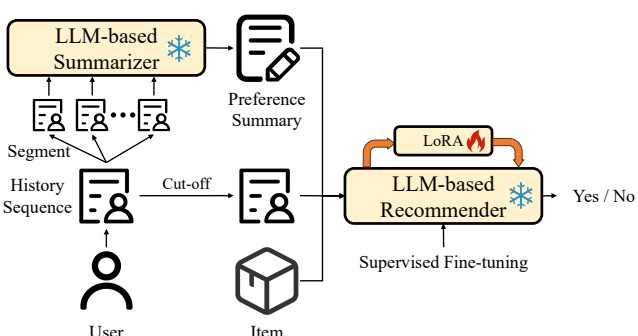

**Figure 1: A schematic diagram of our method. The blue frost symbol indicates fixed parameters, while the red flame symbol signifies parameters that are updated during training.**

(CNN) and Recurrent Neural Network (RNN), we propose two distinct summarization approaches, respectively hierarchical summarization and recurrent summarization. Specifically, for the hierarchical summarization paradigm, we first employ the LLM-based summarizer to extract a summary from each individual block. Subsequently, these individual summaries are concatenated progressively and input to the summarizer again, leading to a higher-level summarization. Through this hierarchical approach, we achieve the final preference summary of the user. For the recurrent summarization paradigm, we initiate the process by using the LLM-based summarizer to extract a summary from the first block. Following this, we iteratively feed the subsequent blocks along with the previously generated summary back into the LLM-based summarizer, prompting it to update the user preference summary based on the new behavioral input. This iterative process continues until the final block, culminating in a comprehensive preference summary of user behavior.

To demonstrate the effectiveness of our method, we conduct extensive experiments on two publicly available datasets from distinct domains, respectively the Amazon-M2 dataset [9] tailored for product recommendation in e-commerce, and the MIND dataset [29] designed for news recommendations on media platforms. The experimental results clearly demonstrate the efficacy of our approach. The major contributions of this paper can be summarized as:

- To the best of our knowledge, we are the first to propose harnessing large language models to address the text-rich sequential recommendation problem.
- We propose to utilize an LLM-based summarizer to encapsulate user behavioral history, and we introduce two distinct summarization paradigms, respectively hierarchical summarization and recurrent summarization.
- We validated the effectiveness of our approach on two open-source datasets from distinct domains. The experimental data and code will be made publicly available upon the acceptance of this paper.

## 2 RELATED WORK

In this section, we will summarize the related works in the following three categories, respectively sequential recommendation, large language models, and LLM for recommendation.

## 2.1 Sequential Recommendation

Sequential recommender systems, often termed as session-based or sequence-aware recommender systems, have garnered substantial attention in recent years due to their significance in modeling user dynamic behaviors and interests. Existing sequential recommendation models primarily employ sequence modeling techniques such as RNN or Transformer to represent user behavior sequences. For example, GRU4Rec [6] proposes to leverage the Gated Recurrent Unit (GRU) model for session-based data modeling. NARM [12] further explores a hybrid encoder with an attention mechanism to capture the user purpose in the current session. BERT4Rec [24] leverages the BERT-based deep bidirectional self-attention architecture to get the representation of user behavior sequences. Furthermore, items in recommender systems might also encompass abundant side information, especially textual data, e.g., the title of news or products. Therefore, some studies have incorporated additional modules for text-rich sequential recommendation scenarios. For example, LSTUR [1] designs a news encoder based on CNN and attention to get the new embedding, and further leverages GRU for sequential modeling. TempRec [28] also designs an item encoder and utilizes Transformer for sequential modeling. However, the aforementioned studies have not extensively explored the utilization of LLMs in sequential recommendation.

## 2.2 Large Language Models

Large Language Models are advanced linguistic models consisting of neural networks ranging from tens of millions to trillions of parameters, trained substantially on vast volumes of untagged texts using methods like self-supervised or semi-supervised learning approaches [16, 33]. The foundation for these LLMs is the Transformer [27] structure, which stands as a cornerstone in the field of deep learning for Natural Language Processing (NLP). Typically, LLMs can be classified into two different types, respectively discriminative LLMs and generative LLMs. For discriminative LLMs, BERT [11] introduces a bidirectional transformer architecture and establishes the concept of the Masked Language Model (MLM) for model pre-training. XLNet [32] incorporates sequence order permutations, facilitating comprehension of word contexts within their surrounding lexical environment. For the generative LLMs, GPT [21] first proposes to pre-train the model by predicting the next word in a sentence. InstructGPT [17] further proposes Reinforcement Learning from Human Feedback (RLHF) for fine-tuning. Llama and Llama-2 [25, 26] are two famous collections of LLMs ranging in scale from 7 billion to 70 billion parameters. In this paper, we select the Llama models as the summarizer and recommender. Recently, several studies have focused on how to extend the input length limitations of existing LLMs [31]. However, the challenges of increased computational overhead and performance degradation remain unresolved. Through the user preference summarization method proposed in this paper, we can handle theoretically infinite user behavior sequences and significantly reduce the training overhead of downstream recommendation models.

## 2.3 LLM for Recommendation

Due to the powerful reasoning capabilities and zero/few-shot learning abilities, LLMs have recently gained significant attention in

the domain of recommender systems. According to the survey paper [30], existing studies on LLM for RS can be divided into two categories, respectively discriminative LLMs for RS and generative LLMs for RS, and the modeling paradigm can be divided into three categories, respectively LLM Embeddings + RS, LLM Tokens + RS, and LLM as RS. Indeed, the discriminative LLM for recommendation mainly refers to the BERT-based models, while the generative LLM for recommendation mainly refers to the GPT-like models. For the discriminative LLMs, U-BERT [20] proposes to utilize the BERT model as embedding backbones and align the representations from the BERT model with the domain-specific data through fine-tuning. For the generative LLMs, since these models have strong zero/few-shot learning abilities, some studies propose to employ these models via prompting methods without fine-tuning. For example, [13] uses ChatGPT as a versatile recommendation model, assessing its performance across five distinct recommendation contexts. Furthermore, several studies propose to further refine the LLMs, aiming to optimize their efficacy. As an illustration, TALLRec[2] suggests enhancing the LLMs via recommendation-focused tuning. In this method, the input derives from user historical patterns, while the output focuses on binary feedback ("yes" or "no"). In this paper, our method mainly utilizes two different generative LLMs for recommendation, combining both of the LLM Tokens + RS paradigm and LLM as RS paradigm. Recently, several studies have also focused on employing LLMs for sequential recommendations [18]. However, they have not considered the difficulties and challenges posed by text-rich user historical behaviors to LLMs.

## 3 PROBLEM FORMULATION

Here we introduce the problem formulation of the text-rich sequential recommendation problem. Given a user $u$, we can first form the historical user behavior sequence of $u$ as $\mathcal{S} = [I_1, \ldots, I_n]$, where $I_i$ is the $i$-th item the user interacted with, e.g., click, buy, read, etc., and $n$ is the length of the user behavior sequence. Each item $I$ has several types of attributes, and can be formulated as $I = [A_1, \ldots, A_m]$, where $A_i$ is the $i$-th type of attribute and $m$ is the total number of attribute types. Furthermore, each attribute $A$ can be formulated in textual form as $A = [w_1, \ldots, w_s]$, where $w_i$ is the $i$-th word and $s$ is the text length. Based on the above, the problem of text-rich sequential recommendation can be formulated as:

DEFINITION 1 (TEXT-RICH SEQUENTIAL RECOMMENDATION). *Given a user u with the corresponding historical user behavior sequence $\mathcal{S}$, and a candidate item $I^c$, the goal of text-rich sequential recommendation is to estimate the click probability of the candidate item for user u, i.e., $g_u : I^c \rightarrow \mathbb{R}$.*

## 4 TECHNICAL DETAILS

In this section, we will introduce our framework in detail. Specifically, we will first introduce how to get the user preference summary by the LLM-based summarizer, including the hierarchical summarization paradigm and the recurrent summarization paradigm. Then, we will elucidate the training process of the LLM-based recommender using the LoRA-based SFT method, and further demonstrate the application of the trained models for recommendation tasks.

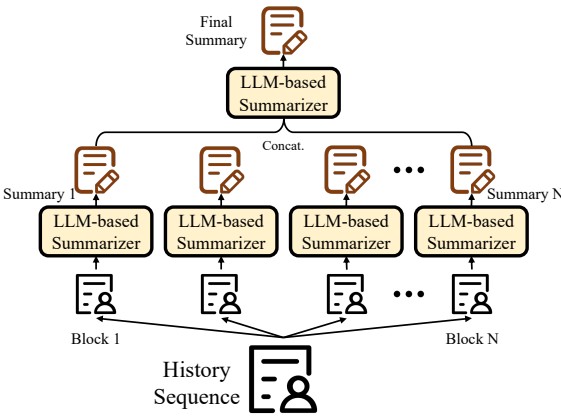

**Figure 2: A schematic diagram of the hierarchical summarization paradigm.**

## 4.1 Hierarchical LLM-based User Preference Summarization

In this section, we will introduce the technical detail of the hierarchical summarization paradigm, which can be shown in Figure 2.

*4.1.1 History Sequence Construction and Segmentation.* To harness the powerful text processing capabilities of LLMs in addressing text-rich sequential recommendation issues, given a user behavior sequence $\mathcal{S} = [I_1, \ldots, I_n]$, we propose to convert $\mathcal{S}$ into a passage of text, which contains information about each item the user has interacted with. However, as we discussed in Section 1, in the context of text-rich sequential recommendation, users may interact with multiple items, each containing extensive textual information. Therefore, directly processing this extensive text with LLMs can present several challenges, such as exceeding the length limitations and excessive computational resource overheads. Therefore, we propose to segment the text, ensuring that each block only contains information related to a few items, making it more manageable for further processing by the LLMs. Specific examples will be presented in the subsequent sections.

*4.1.2 Block Summarization.* In the hierarchical summarization paradigm, after segmenting the text, we will subsequently summarize each text block individually, which allows us to discern the user preference within each specific time frame. For the block summarization, as mentioned in [19], the zero-shot summarization capabilities of LLMs have significantly surpassed the traditional fine-tuned models, and even exceeding human performance. Therefore, in this paper, we employed the Llama-30b-instruct[2] as the summarizer, which is an LLM fine-tuned by instruction-tuning and has achieved commendable results on the Huggingface Open LLM Leaderboard[3]. Moreover, this model offers an appropriate balance between model size and performance. Figure 3 illustrates how we harness the zero-shot summarization capabilities of the LLM by providing a prompt text. In this case, we input an appropriate prompt text according to the required prompt template, asking the model

---

[2]https://huggingface.co/upstage/llama-30b-instruct-2048
[3]https://huggingface.co/spaces/HuggingFaceH4/open_llm_leaderboard

### User:
**Given the historical purchase data of a user, including the titles, descriptions, and attributes of the items they have bought, craft a concise summary that captures the user's preferences, personality, and shopping habits.**

**[Historical Purchase Data]**
**1. Title**: ACEMOOW Bath Mat Non Slip Anti Mould Kids, Extra Long Bathtub Mats with Drain Holes & Suction Cups Machine Washable Rubber Bathroom Shower Mat for Inside Bath Elderly Children - 70 x 35 cm Blue......
......
**5. Title**: Bath Towel Set 100% Cotton (68cm x 137cm - 2 Pack) Premium Bath Sheet, Lightweight and Highly Absorbent Quick Drying Soft Towels for Pool, Spa, and Gym Multipurpose Use–27" x 54" -Fuchsia / Turquoise......

### Assistant:
**This user's historical purchase data indicates that they have an inclination towards non-slip bath mats, luxurious cotton towels, and soft chenille bath rugs. They seem to prioritize quality and durability, as demonstrated by their preference for 100% eco-friendly rubber bath mats, extra thick cotton towels, and heavyweight chenille bath rugs. Their shopping habits suggest that they might value practicality, aesthetic appeal, and overall comfort in their bath products.**

**Figure 3: An example of block summarization on the Amazon-M2 dataset.**

to summarize the user shopping preferences. The model then generates a suitable summary. We can find that, due to the inclusion of fewer items, the summaries obtained in this manner can focus on more intricate details, such as the specific materials of the products. Notably, we can achieve good versatility by modifying the prompt text. For instance, by replacing the shopping-related descriptions in the prompt with news reading-related descriptions, we can utilize the LLM to summarize the news reading preferences.

*4.1.3 Hierarchical Summarization.* Under the hierarchical summarization paradigm, after obtaining the summaries for each block, we propose a hierarchical approach, which involves feeding multiple summaries into the LLM and instructing it to further summarize these summaries, ultimately yielding a comprehensive summary of the user preferences. This process bears a strong resemblance to how Convolutional Neural Network (CNN) extracts higher-level features in a layered manner. Figure 4 illustrates how we design an appropriate prompt to leverage the LLM for this task. We can find that, in contrast to the detailed focus of individual block summaries, the results derived from further summarizing multiple summaries are more abstract and general. They no longer dwell on minutiae but instead capture the overall shopping habits more effectively. This underscores the high level of abstraction and generalization capability that the hierarchical summarization paradigm can offer. It is worth noting that, although in our examples we obtained a final summary of a behavior sequence containing ten items using only two layers of summarization, we can in practice further extend the number of summarization layers, much like adding layers in a convolutional neural network. This theoretically allows us to handle behavior sequences containing information on an infinite number of items.

## 4.2 Recurrent LLM-based User Preference Summarization

In this section, we will introduce the technical detail of the recurrent summarization paradigm, which can be shown in Figure 5.

Inspired by Recurrent Neural Networks (RNNs), the recurrent summarization paradigm operates as follows. After segmenting the user behavior sequence text into blocks, the summary of the first block is extracted. Subsequently, the summary of the preceding block and the user behavior from the next block are input into the LLM-based summarizer to produce an updated summary. This process is iteratively executed until the end of all blocks, resulting in the final user preference summary.

*4.2.1 First Block Summarization.* In the recurrent summarization paradigm, the method for summarizing the first block is essentially consistent with the approach used in the hierarchical summarization paradigm. Figure 6 provides an example of summarizing the first block of a specific user in the MIND dataset.

*4.2.2 Recurrent Summarization.* Having obtained the summary for the first block, we can proceed with a recurrent summarization to derive the final summary of user preferences. Figure 7 demonstrates how we design an appropriate prompt text to harness the LLM for this task. It is evident that we have incorporated more detailed descriptions within the prompt text to ensure the LLM can accurately comprehend the task at hand. The output from the LLM aligns with our expectations, effectively capturing the long-term user interests while updating the summary of their short-term inclinations.

## 4.3 LLM-based Recommendation

After getting the summary of user preferences, we can now employ an LLM-based recommender without concerns about length limitations or excessive computational overhead. We propose to train the LLM-based recommendation model using a Supervised Fine-Tuning (SFT) approach. To be specific, as illustrated in Figure 8, in this paper, we propose constructing a prompt text for the LLM-based recommender system composed of the following five parts:

- **Recommendation Instruction**: Its role is to instruct the LLM to consider both the preference summary and the user recent behaviors to complete the recommendation task. The recommendation task is structured as an output of either "yes" or "no".
- **Preference Summary** This derives from the hierarchical summarization paradigm or recurrent summarization paradigm mentioned earlier, serving to represent the user long-term interests.
- **Recent User Behavior**: This encompasses the items the user has recently interacted with, indicating the user short-term interests.
- **Candidate Item Description**: This offers all textual attributes of the candidate item.
- **Final Answer**: This clarifies whether the user has interacted with the item or not.

Then, we use the following SFT training loss to train the LLM-based recommender as:

$$\mathcal{L}_{sft} = -\sum_{i=1}^{L} \log \Pr(v_i|v_{<i}), \quad (1)$$

where $v_i$ is the $i$-th word in the prompt text and $L$ is the length of the prompt text. The probability $\Pr(v_i|v_{<i})$ is calculated by the LLM

---

**### User:**
**Given a series of user preferences summaries arranged in chronological order, generate a concise summary that encapsulates the user's overall preference. Note that the newly generated summary of user preferences should be consistent with the format of the given summaries.**

**[User Preferences Summaries]**
1. This user's historical purchase data indicates that they have an inclination towards non-slip bath mats, luxurious cotton towels, and soft chenille bath rugs. They seem to prioritize quality and durability, as demonstrated by their preference for 100% eco-friendly rubber bath mats, extra thick cotton towels, and heavyweight chenille bath rugs. Their shopping habits suggest that they might value practicality, aesthetic appeal, and overall comfort in their bath products.

2. The user has purchased a Keplin Non-Slip Bath Mat (Navy) for $9.99 and a NIVEA Care Shower Creme Soft (Almond) 250 ml for $1.0. This suggests the user is detail-oriented, values aesthetics, and invests in homecare and personal products. They showcase preferences towards elegant, high-quality bath mats and mild scented, moisturising personal care products, possibly seeking comfort and hygiene. Their purchasing habits display an inclination towards reputed brands.

**### Assistant:**
**This user is focused on buying high-quality, comfortable, and aesthetically pleasing bath products that prioritize non-slip features and durability. They show a preference for notable brands and carefully select products with a combination of practicality, quality, and comfort in mind, including non-slip bath mats, luxurious cotton towels, and soft chenille bath rugs.**

**Figure 4: An example of hierarchical summarization.**

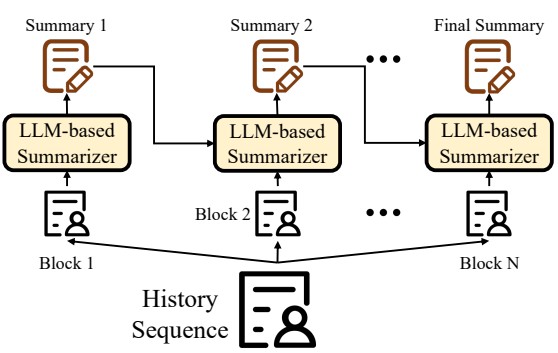

**Figure 5: A schematic diagram of the recurrent summarization paradigm.**

---

**### User:**
**Given the historical news reading data of a user, including the categories and titles of the news, craft a concise summary that captures the user's news reading preferences.**

**[News Reading Data]**
1. **Category**: travel; **SubCategory**: travelnews; **Title**: Southwest Airlines flight diverted: Man assaulted flyers, police say
2. **Category**: sports; **SubCategory**: football_ncaa_videos; **Title**: Who Won Week 7 in College Football?
3. **Category**: video; **SubCategory**: news; **Title**: Man testing new tires flees after crashing into gas station
4. **Category**: travel; **SubCategory**: traveltripideas; **Title**: The Spookiest Place in Every State
5. **Category**: foodanddrink; **SubCategory**: recipes; **Title**: 3 New Alison Roman Recipes You'll Be Making for the Rest of Your Life

**### Assistant:**
**This user has a diverse range of news reading preferences, with interests in travel, sports, video news, and food & drink. Their travel focus includes both travel news and trip ideas, while in sports they are particularly interested in college football. They also enjoy video content and exploring new recipes.**

**Figure 6: An example of the first block summarization on the MIND dataset.**

model following the next-token prediction paradigm. During the training process, we utilize Low-Rank Adaptation (LoRA) [8] for

**Table 1: Statistics of the datasets.**

| description | Amazon-M2 | MIND |
|---|---|---|
| # of different attributes | 10 | 4 |
| # of positive samples in the training set | 10,000 | 10,000 |
| # of positive samples in the validation set | 1,000 | 1,000 |
| # of positive samples in the test set | 1,000 | 1,000 |
| Avg. # of historical user behavior sequence | 13.16 | 16.23 |
| Avg. # of tokens corresponding to an item | 141.45 | 40.83 |

Parameter-Efficient Fine-Tuning (PEFT), which can greatly reduce the number of trainable parameters.

After the training phase is completed, during the testing phase, we remove the "yes" or "no" at the end of the prompt text. We then input this modified prompt $P$ into the large language model and obtain the probabilities predicted by the model for the next word being either "yes" or "no" as:

$$p_{yes} = \Pr('yes'|P), \quad p_{no} = \Pr('no'|P). \quad (2)$$

Finally, we calculate the interaction probability by using the softmax function as:

$$p = \frac{\exp(p_{yes})}{\exp(p_{yes}) + \exp(p_{no})}. \quad (3)$$

## 5 EXPERIMENTS

In this section, we will introduce the datasets used in this paper, the baseline methods, evaluation metrics, and experimental results.

### 5.1 Dataset Description

In this paper, we selected two open-source text-rich sequential recommendation datasets, respectively Amazon-M2 dataset [9] for product recommendation and MIND dataset [29] for news recommendation. We selected records that contain a minimum of 10 and a maximum of 25 interactions. For the training set, we employed a 1:1 ratio for negative sampling, while in the validation set and test set, we utilized a 1:20 ratio for negative sampling. Detailed statistics of these two datasets are shown in Table 1.

### User:
**Given the following summary of a user's news reading preference, and a list of recent news they have read, analyze whether the user's news reading preferences and habits have changed. Taking into account both the existing summary of user preferences and the user's most recent reading records, generate an updated concise summary that captures the user's news reading preferences. Note that the newly generated summary of user preferences should be consistent with the format of the previous Preference Summary. It should serve as a complete summary of the user, rather than a separate narrative of the user's original summary and current preferences.**

[Previous Preference Summary]
This user has a diverse range of news reading preferences, with interests in travel, sports, video news, and food & drink. Their travel focus includes both travel news and trip ideas, while in sports they are particularly interested in college football. They also enjoy video content and exploring new recipes.

[Recent News Reading Data]
1. **Category**: news; **SubCategory**: newscrime; **Title**: Seattle Police officer fires gun while breaking up North Admiral scuffle; detectives investigating
......
5. **Category**: lifestyle; **SubCategory**: lifestylehomeandgarden; **Title**: Genius tiny homes for big families

### Assistant:
**This user maintains diverse news interests, primarily focusing on travel, sports, video news, and food & drink. Their current reading highlights crime stories and college football, with occasional forays into home and garden topics.**

**Figure 7: An example of recurrent summarization.**

### User:
**Given a preference summary of the user, and information related to the historical items the user has purchased, predict whether the user will click on the given next product. Note that the preference summary captures long-term interests, while the recent items indicate short term interests. Both of them should be holistically considered for a more comprehensive understanding of user behavior. Please output 'Yes' or 'No'.**

[Preference Summary]
The user's preferences lean towards non-slip bath mats, luxurious cotton towels, and soft chenille bath rugs, while also valuing quality and durability in their purchases. Their recent purchases include a mix of items, reflecting a balance of comfort, practicality, and elegance. In their latest purchase, the user opted for a versatile, stylish Keplin Non-Slip Bath Mat in Navy color, which is not only water absorbent and quick-drying but also machine washable for a clean and hygienic bathroom. Additionally, they purchased the NIVEA Care Shower Creme Soft, a moisturizing shower body cream enriched with Almond Oil, Vitamins C and E, nourishing skin and providing a mild scent. Their preference summary now consists of a diverse range of products, maintaining an ideal balance of comfort, practicality, and elegance.

[Historical Items]
1. **Title**: Olivia Rocco Bath Mat Plain Super Soft Deep Pile Heavy Weight Micro Bobble Bathmat Bathroom Shower Mat, 50 x 80 cm, Ochre......
......
3. **Title**: NIVEA Care Shower Creme Soft (250 ml) Caring Shower Body Cream Enriched with Almond Oil, Moisturising Shower Gel Body Wash, Skin Moisturiser with Mild Scent......

[Next Item]
**Title**: Creightons Body Bliss Mango & Papaya Bath & Shower (500ml) - Formulated with 90% Naturally Derived Ingredients. 100% Vegan. Cruelty Free. Sustainably Sourced Fruit Extracts......

### Assistant:
**Yes**

**Figure 8: An example of LLM-based recommendation.**

## 5.2 Experimental Settings

*5.2.1 Baseline Methods and Evaluation.* To evaluate the performance of our model for text-rich sequential recommendation, we selected a number of state-of-art methods as baselines. Specifically, we first chose two traditional non-sequential recommendation methods as:

- **NCF [5]**: NCF is a deep learning-based model for collaborative filtering. Max-pooling is used for user representation.
- **DIN [35]**: DIN utilizes attention mechanisms to capture the user interest from the clicked items.

Then, we chose several state-of-the-art sequential recommendation methods as:

- **DIEN [34]**: DIEN adds a sequential modeling part to capture the evolution of user interest compared with the DIN model.
- **GRU4Rec [6]**: GRU4Rec utilizes the GRU model for user behavior sequence modeling.
- **CORE [7]**: CORE uses a linear combination for behavior sequence modeling.
- **NARM [12]**: NARM utilizes RNNs with attention mechanisms for user behavior sequence modeling.
- **SASRec [10]**: SASRec uses self-attention combined with position embeddings for sequence modeling.

Note that for all the above baseline methods, we use the pre-trained BERT [4] for text embedding. Finally, we chose an LLM-based sequential recommendation method as:

Table 2: The performance of different models.

| | Amazon-M2 | | | | | | MIND | | | | | |
| | Recall | | | MRR | | | Recall | | | MRR | | |
| | @3 | @5 | @10 | @3 | @5 | @10 | @3 | @5 | @10 | @3 | @5 | @10 |
|---|---|---|---|---|---|---|---|---|---|---|---|---|
| NCF | 0.8300 | 0.8830 | 0.9440 | 0.7328 | 0.7448 | 0.7529 | 0.7010 | 0.8030 | 0.9240 | 0.5523 | 0.5759 | 0.5926 |
| DIN | 0.7380 | 0.8330 | 0.9240 | 0.5838 | 0.6053 | 0.6174 | 0.7900 | 0.8620 | 0.9330 | 0.6352 | 0.6519 | 0.6616 |
| DIEN | 0.7330 | 0.8170 | 0.9070 | 0.5922 | 0.6114 | 0.6229 | 0.7300 | 0.8200 | 0.9140 | 0.6045 | 0.6251 | 0.6379 |
| GRU4RecText | 0.4420 | 0.5590 | 0.7350 | 0.3355 | 0.3621 | 0.3855 | 0.6650 | 0.7970 | 0.9260 | 0.5305 | 0.5610 | 0.5787 |
| NARMText | 0.8410 | 0.8860 | 0.9330 | 0.7475 | 0.7577 | 0.7638 | 0.5820 | 0.7330 | 0.8930 | 0.4142 | 0.4489 | 0.4703 |
| SASRec | 0.6550 | 0.7570 | 0.9040 | 0.4938 | 0.5173 | 0.5374 | 0.8420 | 0.8960 | 0.9410 | 0.7447 | 0.7574 | 0.7636 |
| CORE | 0.5230 | 0.4632 | 0.6450 | 0.4527 | 0.4632 | 0.4728 | 0.5170 | 0.5580 | 0.6370 | 0.4392 | 0.4488 | 0.4586 |
| TALLRec | 0.8790 | 0.9050 | 0.9460 | 0.8585 | 0.8644 | 0.8697 | 0.8580 | 0.9020 | 0.9590 | 0.7708 | 0.7807 | 0.7885 |
| LLM-TRSR-Hierarchical | **0.8910** | 0.9120 | 0.9490 | 0.8597 | 0.8643 | 0.8693 | **0.9160** | **0.9430** | 0.9750 | **0.8505** | **0.8568** | **0.8611** |
| LLM-TRSR-Recurrent | **0.8910** | **0.9130** | **0.9570** | **0.8632** | **0.8681** | **0.8737** | 0.9060 | 0.9390 | **0.9840** | 0.8400 | 0.8475 | 0.8534 |

- TALLRec [2]: TALLRec proposes to leverage LLMs for recommendation by instruction tuning.

To evaluate the performance of different models, we selected Recall@K and Mean Reciprocal Rank (MRR)@K as evaluation metrics, where the value of K can be 3, 5, and 10.

*5.2.2 Implementation Details.* We conducted experiments using a cluster composed of 12 Linux servers, each equipped with 8*A800 80GB GPUs. We selected Llama-30b-instruct[4] with 8-bit quantization as the summarizer and Llama-2-7b[5] with BF16 as the recommender. We used PyTorch[6] and TRL[7] library for the SFT step and we used LoRA with the rank equal to 8. We used the AdamW [15] optimizer with learning rate as 1e-4 and batch size as 1 for SFT, and we set gradient accumulation steps as 64 and epoch number as 8. We also used Deepspeed [23] with ZeRO stage as 2 for distributed training. Furthermore, we set the max length of tokens of LLMs as 2048 and the item number in a block as 5.

## 5.3 Overall Performance

To demonstrate the effectiveness of our model on reciprocal recommendation, we compare LLM-TRSR with all the baseline methods, and the results are shown in Table 2. Note that we set the number of historical items in the prompt text for recommendation as 3, and the suffix '-Hierarchical' or '-Recurrent' indicate the paradigm through which user preference summaries are obtained. From the results, we can get the following observations:

(1) The performance of our model surpasses all of the baseline methods on different evaluation metrics and different datasets. This clearly proves the effectiveness of our LLM-TRSR model for text-rich sequential recommendation.

(2) Recommendation approaches based on LLMs consistently outperform traditional methods, underscoring the substantial potential of LLMs in the realm of sequential recommender systems.

---

[4]https://huggingface.co/upstage/llama-30b-instruct-2048
[5]https://huggingface.co/meta-llama/Llama-2-7b-hf
[6]https://pytorch.org/
[7]https://huggingface.co/docs/trl/index

(3) On the Amazon-M2 dataset, LLM-TRSR-Recurrent outperforms LLM-TRSR-Hierarchical, while the opposite holds true for the MIND dataset. This suggests that different paradigms for summarizing user preferences might be suitable for varying scenarios. For instance, the recurrent paradigm may capture the user preference transitions more effectively, whereas the hierarchical paradigm might better capture the user overarching interests.

## 5.4 Discussion on Historical Item Number

In Section 4.3, we mentioned that the prompt text fed into the LLM-based recommender includes information about items the user has historically interacted with, and in Section 5.3 we set this number as 3. In this section, we will explore the impact of varying numbers of historical items on the results, and the results are shown in Figure 9. We can find that as the number increases, the model performance initially improves and then declines, with the optimal performance occurring when the number is set to 3. This suggests that either too few or too many historical items are not conducive to enhancing the model performance. Additionally, we note that even when the number of historical items is set to 0, meaning the model recommends solely based on user preference summaries, it still achieves reasonably good performance. This underscores the effectiveness of our proposed summarization methods.

## 5.5 Discussion on Parameter Size

It is well-known that the parameter size of LLMs can significantly impact their performance. In this section, we will discuss the influence of parameter size on the framework proposed in this paper.

*5.5.1 Discussion on Parameter Size of Recommender.* In this paper, we selected Llama-2-7b as the backbone model of the LLM-based recommender. To investigate the performance of recommender with varying parameter sizes, we selected models from Pythia [3], a suite of 16 LLMs with different sizes, as comparisons. Specifically, we selected two smaller-scale models, recpectively Pythia-1.4b[8] and Pythia-2.8b[9], and used them to replace the Llama-2-7b model. The

---

[8]https://huggingface.co/EleutherAI/pythia-1.4b
[9]https://huggingface.co/EleutherAI/pythia-2.8b

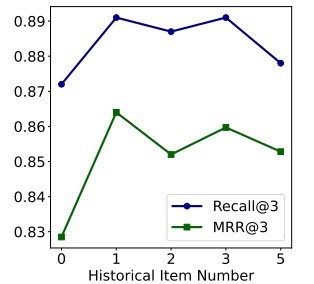 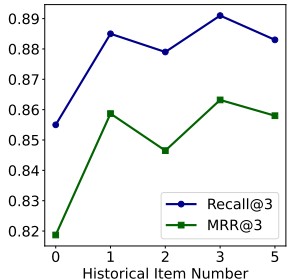

(a) The performance of LLM-TRSR-Hierarchical with different historical item number on the Amazon-M2 dataset.

(b) The performance of LLM-TRSR-Recurrent with different historical item number on the Amazon-M2 dataset.

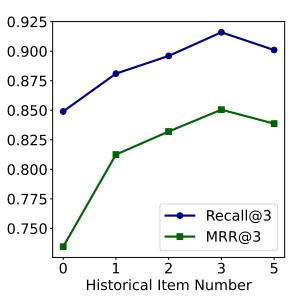 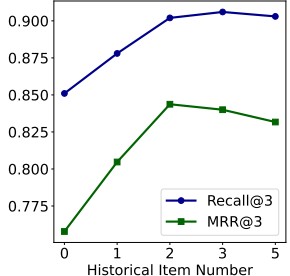

(c) The performance of LLM-TRSR-Hierarchical with different historical item number on the MIND dataset.

(d) The performance of LLM-TRSR-Recurrent with different historical item number on the MIND dataset.

**Figure 9: The performance of different models with different historical item number on different datasets.**

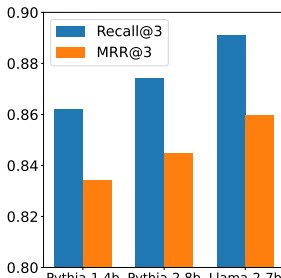 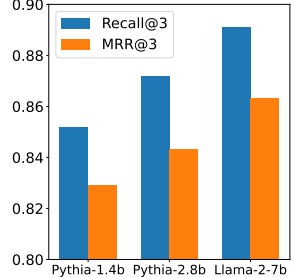

(a) The performance of recommenders with hierarchical summarization paradigm on the Amazon-M2 dataset.

(b) The performance of recommenders with recurrent summarization paradigm on the Amazon-M2 dataset.

**Figure 10: The performance of recommenders with different parameter scales on the Amazon-M2 dataset.**

results are presented in Figure 10. From the results, we observe that models with a larger scale generally achieve better performance. However, note that larger models also demand greater computational resources. Thus, in practical application scenarios, striking a balance between model performance and computational overhead is a matter worth considering.

*5.5.2 Discussion on Parameter Size of Summarizer.* In this paper, we employed the Llama-30b-instruct model as the summarizer,

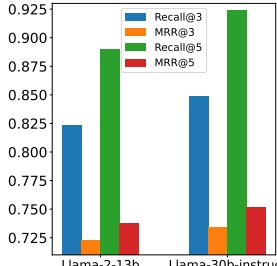 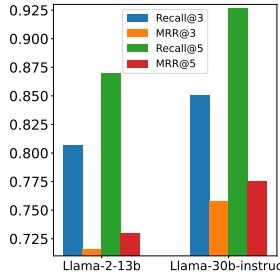

(a) The performance of summarizers with hierarchical summarization paradigm on the MIND dataset.

(b) The performance of summarizers with recurrent summarization paradigm on the MIND dataset.

**Figure 11: The performance of summarizers with different parameter scales on the MIND dataset.**

leveraging its zero-shot summarization capabilities for user preferences summarization. To investigate the differential capabilities of LLMs of varying scales in summarizing user preferences, we experimented with Llama-2-13b[10] as the summarizer. Furthermore, to accentuate the summarization capabilities of different models, we set the historical item number in the recommendation prompt as 0. The experimental results are shown in Figure 11. From the results, we can find that the summarization capability of Llama-30b-instruct significantly surpasses that of Llama-2-13b. Additionally, we observed that the summaries generated by Llama-2-13b were of inferior quality. Both the content and format were disorganized, making them difficult for humans to comprehend. This suggests that only LLMs with a substantial number of parameters can proficiently perform the task of zero-shot user preference summarization.

## 6 CONCLUSION

In this paper, we investigated the application of Large Language Models for Text-Rich Sequential Recommendation (LLM-TRSR). Specifically, we first proposed segmenting the user behavior sequences. Then, leveraging the zero-shot summarization capabilities of large language models, we employed an LLM-based summarizer to encapsulate user preferences. Notably, we introduced two distinct preference summarization paradigms, respectively hierarchical summarization and recurrent summarization. Subsequently, we proposed to use an LLM-based recommender for sequential recommendation tasks, with parameters being fine-tuned using Supervised Fine-Tuning (SFT). Low-Rank Adaptation (LoRA) was also utilized for Parameter-Efficient Fine-Tuning (PEFT). Experiments conducted on two public datasets compellingly evidenced the efficacy of our approach proposed in this paper. Additionally, we discussed the impact of different parameter scales of LLMs on the experimental results.

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
