# OpenReview forum: "Harnessing Large Language Models for Text-Rich Sequential Recommendation"
_ACM.org/TheWebConf/2024/Conference — TheWebConf24_

### Official Review · Reviewer_FN4V · 2023-11-05

**Novelty:** 4
**Technical Quality:** 4

**Review:**

This manuscript discusses the challenges faced by Large Language Models (LLMs) in text-rich sequential recommendation scenarios, where items contain rich textual information. The limitations of LLMs in terms of input length, computational overhead, and capturing shifts in user preferences are highlighted. To address these challenges, the document proposes a novel framework called LLM-TRSR (Large Language Models for Text-Rich Sequential Recommendation). The framework includes two unique summarization techniques: hierarchical summarization and recurrent summarization. It also incorporates Supervised Fine-Tuning (SFT) and Parameter-Efficient Fine-Tuning (PEFT) methods. Experimental results on two public datasets demonstrate the effectiveness of the approach.

Strengths:
1. Proposal of harnessing LLMs for text-rich sequential recommendation.
2. Introduction of hierarchical summarization and recurrent summarization techniques.

Weakness:
1. The authors do not provide the ablation study results to demonstrate the effectiveness of the proposed modules.
2. The authors do not provide comparative experimental results on the number of parameter and time efficiency.

**Questions:**

1. Could you provide the ablation study results to demonstrate the effectiveness of the proposed modules?
2. Could you provide comparative experimental results on the number of parameter and time efficiency?
3. Why the authors only utilize RNN like: GRU to model the sequential information and do not take the transformer into account?

**Reviewer Confidence:**

3: The reviewer is confident but not certain that the evaluation is correct

**Scope:**

4: The work is relevant to the Web and to the track, and is of broad interest to the community

---

### Official Review · Reviewer_iWHz · 2023-11-23

**Novelty:** 3
**Technical Quality:** 4

**Review:**

The authors propose to use the powerful comprehension ability of large language models to summarize the user preferences. To handle long user sequence, they design a hierarchical summarization approach. Finally, they use LLMs as recommenders by supervised finetuning strategy, with all generated preference data and original history list as input. They test their LLM-TRSR model with the Amazon and MIND dataset for product and news recommendation respectively.

Pros:

- The authors propose a two-stage framework that use LLMs for both feature enhancement and recommendation. The idea of hierarchical summarization is simple but seems good.
- Experiments are conducted on two types of the datasets with various models as baselines.
- The authors put many examples in the article, making the paper easy to follow.


Cons:

- Some relevant papers are missing mentions. [1][2][3]
- More ablation studies required. The author mainly test their two-stage framework entirely with other baselines. However, each stage is not separately tested. For example, 1) can the generated text be used for base models such as GRU4RecText? 2) how heavy will the performance drops if not using the generated summarization during LLM-based recommendation tuning? 3) Other work such as ONCE [3] also proposes to use LLM for item-level and user-level summarization and the MIND dataset is tested in their paper. Can you compare the summarization quality of LLM-TRSR and ONCE?
- More baselines required. It would be better if using text-based models as baselines for text-based dataset. For example, NAML[4], NRMS[5], LSTUR (reference 1 in your paper) are typical models in news recommendation. It seems unfair to compare text-based methods with id-based ones.

[1] Harte, Jesse, et al. "Leveraging Large Language Models for Sequential Recommendation." Proceedings of the 17th ACM Conference on Recommender Systems. 2023.
[2] Liu, Junling, et al. "Llmrec: Benchmarking large language models on recommendation task." arXiv preprint arXiv:2308.12241 (2023).
[3] Liu, Qijiong, et al. "ONCE: Boosting Content-based Recommendation with Both Open- and Closed-source Large Language Models." arXiv preprint arXiv:2305.06566 (2023).
[4] Wu, Chuhan, et al. "Neural news recommendation with attentive multi-view learning." arXiv preprint arXiv:1907.05576 (2019).
[5] Wu, Chuhan, et al. "Neural news recommendation with multi-head self-attention." Proceedings of the 2019 conference on empirical methods in natural language processing and the 9th international joint conference on natural language processing (EMNLP-IJCNLP). 2019.

**Questions:**

1. Can you add more relevant papers in your related work section?


2. Can you make deep ablation study to verify the effectiveness of each stage? More details please refer to the second cons.

3. Can you explain the reason of using current baseline set? Can you add more text-based models as baselines as mentioned in the thrid cons.

**Ethics Review Description:**

/

**Reviewer Confidence:**

4: The reviewer is certain that the evaluation is correct and very familiar with the relevant literature

**Scope:**

4: The work is relevant to the Web and to the track, and is of broad interest to the community

---

### Official Review · Reviewer_4ptF · 2023-11-23

**Novelty:** 4
**Technical Quality:** 5

**Review:**

Pros:

1.This paper addresses the long-text problem that incorporating contextual text information for sequential recommendation will result in over-length problem and suboptimal performance, which is practical for text-based sequential recommendation.

2.The paper proposes to split long user sequences into blocks and summarize blocks for prediction, which proves to be effective,

3.An extensive study for user behavior modelling paradigms, including recurrent and hierachical summarization techniques, are conducted.

4.SFT and LoRA techniques are used for further performance boost.

Cons:

1.Potential efficiency problem is not discussed. Not sure if the proposed method is applicable for real-world recommender systems.

2.Expeirments on larger datasets is expected, where conventional recommender systems may be more powerful. I would like to see the performance gap on larger datasets.

**Questions:**

1. How is the efficiency of the proposed method?
2. Detailed statistics of the used datasets and sequence/token lengths will be expected.
3. How is the performance compared with the retrieval-based method for long-text LLM-based recommendation model? For example, ReLLa.

**Reviewer Confidence:**

4: The reviewer is certain that the evaluation is correct and very familiar with the relevant literature

**Scope:**

4: The work is relevant to the Web and to the track, and is of broad interest to the community

---

### Official Review · Reviewer_zu8S · 2023-11-28

**Novelty:** 2
**Technical Quality:** 6

**Review:**

Summary:

In this paper, the authors propose an LLM-based algorithm for the text-rich sequential recommendation problem. To put it simply, the problem is to come up with the probability distribution/score for Prob(next_item = item | context), where context is all the historical interactions of the user (purchases, clicks, etc.). Also each item is represented by large amounts of text (attributes, titles, descriptions). The key challenge in this problem is that the context length is very high, and won’t fit in an LLM. (Most LLMs only handle up to 4K tokens). To solve this problem, the authors propose two summarization strategies, one hierarchical and the other recurrent in order to compress the context. Once we have a context that fits within the LLM input, the authors LORA-tune another LLM to predict the next item.


Strong:
* The paper is well written and a pleasure to read. Good coverage on the related work. Also, thanks for providing much of the parameters required to reproduce such a model.


Weak:
* The novelty of the proposed approach is limited. The key contribution, which is that of compressing the context, is a well studied problem in the text summarization literature.

* Dataset used: minimum of 10 interactions and a maximum of 25 interactions. In some sense, it is a pretty “easy” dataset. It would be interesting to see if using LLMs can be used to solve the harder zero-shot / cold-start problems.

* Most baselines are weak since they do not use all the features that the proposed model uses. For instance, the collaborative filtering models only use the interaction, but not the text. Also most of the BERT based models only use 0.345 B parameters (assuming BERT large) whereas the proposed models use LLaMa (at least 7B params or 30B params). The only “real” comparison is TallRec.

Notes:

* Eqn (3): we already started with probabilities (not logits), no need to use another exp.

* Please combine Figure 10 with Table 2. That way, it is easy to compare the performance of models with similar numbers of parameters. If possible try to use even smaller models more similar to the size of BERT.

* Understanding why the “recurrent” summary is better than the “hierarchical” summary seems like an interesting problem that the authors might want to elaborate on. It seems to me that the recurrent summary might naturally pay more attention to the more recent interactions, and thereby be more important to the next item prediction. It will be interesting to verify such a hypothesis.

**Questions:**

I don't have any questions for the authors, but a few suggestions to improve the paper:

* Eqn (3): we already started with probabilities (not logits), no need to use another exp.

* Please combine Figure 10 with Table 2. That way, it is easy to compare the performance of models with similar numbers of parameters. If possible try to use even smaller models more similar to the size of BERT.

* Understanding why the “recurrent” summary is better than the “hierarchical” summary seems like an interesting problem that the authors might want to elaborate on. It seems to me that the recurrent summary might naturally pay more attention to the more recent interactions, and thereby be more important to the next item prediction. It will be interesting to verify such a hypothesis.

**Ethics Review Description:**

No concerns

**Reviewer Confidence:**

4: The reviewer is certain that the evaluation is correct and very familiar with the relevant literature

**Scope:**

3: The work is somewhat relevant to the Web and to the track, and is of narrow interest to a sub-community

---

### Decision · Program_Chairs · 2024-01-22

**Decision:**

Accept

**Comment:**

This work tries to address the long text problem in sequential recommendation by leveraging the power of large language models to summarize text. The paper is well written and easy to read. The evaluation could still be improved.